# Predicting the Durability of Solid Fired Bricks Using NDT Electroacoustic Methods

**DOI:** 10.3390/ma15175882

**Published:** 2022-08-25

**Authors:** Vojtěch Bartoň, Richard Dvořák, Petr Cikrle, Jaroslav Šnédar

**Affiliations:** 1Institute of Building Testing, Faculty of Civil Engineering, Brno University of Technology, Veveří 331/95, 602 00 Brno, Czech Republic; 2Institute of Physics, Faculty of Civil Engineering, Brno University of Technology, Veveří 331/95, 602 00 Brno, Czech Republic

**Keywords:** solid fired brick, defects in the internal structure, non-destructive testing, resonant pulse method, material durability, machine learning

## Abstract

Historical buildings and monuments are largely made of brickwork. These buildings form the historical and artistic character of cities, and how we look after them is a reflection of our society. When assessing ceramic products, great emphasis is placed on their mechanical properties, whilst their durability is often neglected. However, the durability or resistance to weathering of masonry elements is just as important as their mechanical properties. Therefore, this work deals with predicting the durability of solid-fired bricks before they are used when reconstructing monuments and historical buildings. Durability prediction is assessed by identifying defects in the material’s internal structure. These faults may not be visible on the element’s surface and are difficult to detect. For this purpose, non-destructive electroacoustic methods, such as the resonant pulse method or the ultrasonic pulse method, were used. Based on an analysis of the initial and residual mechanical properties after freezing cycles, four durability classes of solid-fired bricks were determined. This work aimed to find a way to predict the durability (lifetime) of an anonymous solid-fired brick, expressed in terms of the number of freeze cycles the brick would last, based on non-destructive measurements.

## 1. Introduction

Historical buildings and monuments are our real wealth. These buildings represent the foundation of our cultural heritage and are the most tangible legacy of our past. The management of monuments is a reflection of society’s cultural advancement. An integral part of this reflection is not only the care of historical objects themselves but also the effort to find the most appropriate way to preserve them [1].

Building materials are often assessed primarily in terms of mechanical properties, such as compressive strength or flexural tensile strength [2,3]. In terms of the durability of masonry elements, the frost resistance test, or the ability of the material to withstand repeated freezing and thawing, may be specified [4]. However, these tests are usually destructive or semi-destructive, which is not desirable, especially in the case of monuments. The question of the brickwork’s durability is still relegated to the background. However, the durability or the ability of the material to resist external influences (weathering, chemical corrosion, or salt crystallisation) is an equally important factor for a material such as solid-fired bricks. Especially when the brickwork is not protected from the weather by plaster or any other surface treatment. Such masonry forms the visual part of the buildings and is the most susceptible to weather damage. Despite the initial neglect of this issue, the durability of building materials has already been addressed by a number of authors. However, the attention of authors is often directed to the durability of concrete structures [5]. In the case of the authors’ focus on the durability of masonry buildings, it is more of an attempt to extend the durability of structures by understanding the defects already present in masonry buildings and the procedures for reconstruction (restoration) [6,7,8] or [9] but not the prevention of such defects. Other authors deal with the durability of solid-fired bricks (hereinafter SFB) and the effects that reduce durability; however, this is often a destructive way and the element cannot be subsequently used in historical construction [10,11]. Particularly beneficial to this research were the works of authors dealing with the issue of frost damage to porous materials, which gave a better picture of the effect of the internal structure on the durability of masonry elements [7,12,13,14].

The effort to understand the influences that reduce the durability of solid brick and the effort to find the most effective ways to renovate existing structures without major interventions is undoubtedly very beneficial. However, there is often a situation where such buildings cannot be reconstructed without replacing (substituting) individual wall elements. In this case, the question arises whether to reconstruct the elements with historical bricks or to use masonry elements made with modern technologies. Both approaches have their advantages and disadvantages.

The main advantage of using modern wall elements lies in the assumption of less variability in their mechanical properties (especially compressive strength or flexural strength), a uniform appearance as well as durability. It is possible to ensure a stable technological process during the commissioning of modern wall elements and adjust it if necessary. However, the emphasis in monuments is often on preserving the authenticity of the brickwork, which is often problematic when modern masonry elements are used (Figure 1a). The main advantage of reusing historic masonry elements (so-called “upcycling”) is that the authenticity of the brickwork is preserved (Figure 1b).

In recent years, the issue of sustainable development has also increasingly come to the fore, and the use of historic wall elements takes on this approach. This is a more environmentally friendly method, as there is no need to produce new masonry elements and the waste from demolished brick buildings is minimised [15].

However, in the case of using historic solid bricks for reconstructing masonry buildings, it is necessary to take into account the great variability of individual elements. Even in the case of taking historic bricks from a single demolished building, the uniformity of the solid bricks is not guaranteed. This is particularly evident during the construction of larger buildings, i.e., masonry elements were supplied from several brickworks at once. It must also be assumed that these buildings have been reconstructed in the past and therefore contain masonry elements from different time periods; this issue has been addressed in the past by authors, e.g., [16]. For these reasons, it is quite difficult to obtain a sufficient quantity of bricks of similar appearance and properties, and it is particularly difficult to demonstrate the requisite quality. When upcycling historic bricks for heritage buildings, it is necessary to consider a number of factors that may affect their quality.

One of the basic factors is the material that was used to produce the historic wall elements. In particular, the type of clay used (ferruginous or loamy). However, it cannot be said that masonry elements made of the same material in the same brickworks will have identical properties or appearances. The quality of the bricks varied even within the same kiln, mainly depending on the bricks’ location in the kiln (different firing temperatures). The firing temperature can significantly affect the mechanical properties, colour, dimensions or, for example, the absorption of the elements.

If the firing temperature is too high, the element usually has a darker firing colour (sometimes up to purple), higher compressive strength and significantly lower absorption. However, if the firing temperature is too high, shrinkage of the elements and deformation of edge flatness often occurs (Figure 2a). Conversely, at low firing temperatures, the element typically has a light brown (orange) firing colour, lower compressive strengths, and significantly higher absorption (Figure 2b). The marking of the elements "A5" and "12" is only to simplify the research.

In terms of the durability of masonry elements, increased water absorption is the most challenging, as it is closely related to resistance to the effects of repeated freezing and thawing [17].

Another factor that significantly affects the durability of masonry elements is their defects. The defects in the elements may have already occurred during the production of the wall elements. Delamination of the individual layers can be mentioned as a defect already occurring during production. When using screw presses, this is a defect in the elements caused by the rotational movement of the mixture used or, in the case of handmade bricks, delamination caused by poor mixing of the clay mass. It mostly manifests only as a result of freezing cycles, when it takes the form of concentric spiral cracks along which the brick gradually crumbles. This type of defect is not only typical of historic bricks but also occurs in abundance in newly manufactured solid-fired bricks.

Other defects in the wall elements may arise due to their history. In particular, when repeated freezing and thawing has caused cracks, chips or microstructural damage to the elements. The main indicator may be the location of the bricks in the past. If the element has been exposed to climatic influences for many years, it can be expected that these defects will occur more frequently. In addition to damage caused by climatic influences acting on the elements, mechanical damage (breaking off of a corner, etc.) may also occur.

When assessing the extent of defects in individual elements, it is important to note that a large proportion of defects are not visible on the surface of the elements. These defects in the internal structure of the elements significantly reduce their durability. These defects are usually caused by increased humidity, when water freezes in winter and increases its volume, causing stresses in the elements. The development of defects in the internal structure need not be linked to the development of defects visible on the surface. The authors of the paper have dealt with this issue in the past. In the framework of [18] 3 types of defect development have been defined:Both surface defects and defects in the internal structure of the element develop;only surface defects develop; the development of defects in the internal structure is negligible;the development of surface defects is indistinct; only defects in the internal structure develop.

The most problematic of the above types of pattern damage is the third case. This is because, in this case, only the internal structure is damaged without damaging the surface of the test sample (Figure 3). Visually, the element may appear free of defects even after a number of freeze cycles, but defects in the internal structure significantly reduce both the mechanical properties of the elements and their durability.

The reconstruction of monuments with historic solid bricks can therefore be a challenging discipline, particularly because of the great variability and difficulty in monitoring the quality of the elements. Due to the great variety of historical masonry elements, their quality cannot be guaranteed based on standard tests (on 5 or 10 bricks). For this reason, non-destructive methods are proposed that allow many more bricks to be tested, and these can subsequently be incorporated into the structure.

For these reasons, completely non-destructive electroacoustic methods, namely the resonance method and the ultrasonic method, were chosen to predict the masonry elements’ durability or resistance to weather conditions (mainly repeated freezing and thawing). Based on the identification of defects in the internal structure of the elements, a total of 4 durability classes were determined with the recommended use of the wall elements. The individual methods and procedures were chosen so they can be applied in practice while ensuring the highest possible reliability. The problem of detecting defects in materials by electroacoustic methods has been addressed in the past by a number of authors, e.g., [5], but here the problem of detecting defects in the material caused by loading (mechanical failure) is addressed.

This work presents an innovative way of using non-destructive resonance and ultrasound methods. Non-destructive methods are commonly used in civil engineering today. However, these methods are mainly focused on mechanical properties. These electroacoustic methods are also used for a kind of identification of defects in the internal structure. However, they are generally applied to concrete structures and there are no clearly defined parameters for detecting defects in the internal structure. This paper deals not only with the actual use of these methods but also with the definition of the parameters that can be used to detect these defects (obtained from the spectrum of natural frequencies) and to predict the durability of the material based on them. Prediction of durability was determined by statistical analysis of a series of experimental measurements. Currently, the method used in-situ is testing fired solid bricks by impacting them with a steel hammer. The evaluation of the quality (durability) of the elements depends only on the individual judgement of the worker based on the acoustic response. Thus, the proposed method is a significant improvement, where a human factor is reduced, and a classification model and algorithm of machine learning are being used.

## 2. Materials and Methods

This work aims to determine the durability criteria of historic, solid-fired bricks before they are used when reconstructing monuments and historic buildings. The elements must not be damaged in any way after their durability has been assessed so that they can be incorporated into the structure. For this reason, completely non-destructive electroacoustic methods, such as the resonance method and the ultrasonic pulse method, were used. A balance had to be struck between the complexity of the criteria to be assessed and the greatest possible accuracy of the results. The durability of the masonry elements is assessed by identifying defects in their internal structure. To identify these defects, spectra of the first natural frequencies determined by the resonance method were analysed.

Every rigid material is subject to vibration due to an external impulse. The phenomenon, when the frequency of this introduced impulse is identical to the natural frequency of the element, is called resonance. This phenomenon is exploited by the resonant pulse method. The undeniable advantage of this method is that, in addition to the first natural frequencies, it was possible to record the entire spectrum of frequencies in the chosen range. This fact has been used in this work, where the defects in the internal structure of the elements are identified by analysing the spectrum of the first natural frequencies and then the durability of the masonry elements is predicted. There are countless types of vibrations that can be induced in rigid bodies. As a rule, however, three types of these oscillations are used, namely:first natural frequency of longitudinal oscillation—f_L_,first natural frequency of the torsional oscillation—f_T_,first natural frequency of transverse oscillation—f_F_ [19].

In this study, only the first two types of oscillations (longitudinal and torsional) were used. The defects in the internal structure of the elements that appear during transverse vibrations are also largely reflected in the frequency spectrum of the torsional vibrations. An example of determining the first natural frequencies is shown in Figure 4a. A figure of the location of the exciter (hammer strike) and a sensor for each type of oscillation are shown in Figure 4b. Handyscope HS3 (oscilloscope) with a piezoceramic sensor was used to determine the natural frequencies of the test elements.

Another method for detecting defects in the internal structure of the elements was the ultrasonic pulse method, specifically the pundit PL-200 from Screening Eagle (formerly Proceq). The principle of the ultrasonic pulse method is based on the mechanical wave of particles through the environment. In practice, the ultrasonic pulse method is used mainly for determining the uniformity of concrete structures and determining the deformation properties of the material, and, to a lesser extent, for determining the compressive strength of concrete structures. This method can also be used to detect defects in the internal structure [20]. For the purpose of this study, the passage times of the direct sounding in the transverse (T_T_) and longitudinal (T_L_) directions of the element were measured in three lines (Figure 5a). An example of ultrasonic wave transit times is shown in Figure 5b.

Prior to conducting laboratory tests to determine the durability criteria for historic solid brick, the applicability of the above methods was verified directly during the reconstruction of a historic brick building. The authors aimed to set the durability criteria for solid bricks so that the durability could be evaluated on-site, if necessary. The authors had a unique opportunity to test the methods on a baroque brick bridge near the village of Mikulov (close to the Czech-Austrian border). This brick bridge is unique for several reasons. Most of the preserved bridges from this period are made of stone masonry. In the case of this bridge, stone masonry was used only at the foot of the piers; the rest of the bridge is made of brickwork. Another unique feature of this bridge is its length; this fifteen-arch bridge is 100 m long. The bridge had been in disrepair for many years, was overgrown with grass and covered with mud deposits. The hope of saving the bridge was raised in 2016 when the project for its restoration began. The reconstruction of the bridge started in 2018 and took 2 years. For its reconstruction, it was decided to “upcycle” the historic wall elements [21,22]. The opening ceremony of the bridge took place in October 2020 and the reconstruction was awarded “Monument of the Year” of the Czech Republic in 2021.

During the reconstruction of the bridge, a series of measurements on the historic in-situ fired bricks were made using the resonance method and the ultrasonic pulse method. Based on these measurements and basic assumptions, wall elements of low quality and wall elements with assumptions for very good quality and durability were selected.

The selected elements were then subjected to a freeze resistance test in the laboratory and then the mechanical properties of the elements were determined. Laboratory tests (determination of frost resistance, flexural tensile strength and compressive strength) confirmed the initial estimates of durability and the quality of the masonry elements made by in-situ non-destructive methods. These results have been published in [23].

The ultrasonic pulse method used during the reconstruction of the bridge did not yield significant results, whereas the resonant pulse method proved to be very promising.

Once the applicability of the in-situ methods had been verified during the reconstruction of the heritage building, the main work to establish the durability criteria for historic solid bricks could begin. The experiment itself is inherently simple. The principle of determining the durability criteria is based on an analysis of the spectrum of the first natural frequencies and their change in the saturated state compared to the dried state. However, a relatively large number of test samples was required. Thus, a total of 41 historic bricks were selected. The test specimens were taken from various demolished buildings over a period of about three years, varying in age (from the Gothic, Baroque and Renaissance periods), size, appearance and expected quality, so that the whole range of test bodies was covered. The test elements were first thoroughly cleaned, and their dimensions were determined. They were subsequently dried to a steady state weight, i.e., a condition where their weight did not change by more than 0.2% during 24 h of drying at 105 °C. For all samples, their dry weights were determined, and any surface defects of the elements were carefully recorded.

The first natural frequencies of the longitudinal *f_L,V_*_0_ and torsional *f_T,V_*_0_ oscillations were determined using the resonant pulse method. Subsequently, the elements were completely saturated with water. The elements were immersed in distilled water so that they did not touch. The water was then brought to a boil, which was maintained for a further 5 h. The elements were left in the water for a further 16 h; this post corresponds to [24]. In the fully saturated state, the mass and first natural frequencies of the elements were again determined using the resonance method *f_L,NV_*_0_ in the longitudinal direction and *f_T,NV_*_0_ in the transverse direction.

The masonry elements were divided into a total of 4 durability classes depending on the number of freezing cycles the bricks could withstand. One F-T cycle consists of 16 h of freezing (at −20 °C) and 2 h of thawing in water (at 15 ÷ 30 °C) in an automatic freezer [4]. At the same time, the possible use of elements from each group was defined:1st class—bricks usable in exposed outdoor environments (uncovered ledges, places above ground with rising damp, etc.2nd class—bricks usable in outdoor environments, less exposed areas (e.g., vertical surface masonry except for plinths, masonry infill) or in indoor exposed areas (e.g., wine cellars)3rd class—bricks usable only indoors, in a dry environment4th class—bricks not suitable for reuse.

Input measurements taken prior to the freezing cycles were used to develop the shelf-life prediction model. For the evaluation of the frequency spectra from the resonance method, a feature extraction method was used to obtain the key characteristic parameters of the spectra under consideration. This method is commonly used during the dimensionality reduction of large datasets [25] and is widely used in the prediction of the lifetime of structures in both civil and mechanical engineering. Thus, in addition to the dominant resonant frequency, other parameters such as amplitude, peak width at mid-peak prominence, and peak prominence were extracted from all spectra. From each spectrum, the following parameters of the first three dominant peaks were selected in descending order of prominence. An example of these characteristics is shown in Figure 6 shows the recorded signal.

Figure 7 shows the frequency spectrum with the peaks and their characteristics highlighted.

A similar approach in the evaluation of the resonance method can be found in the foreign literature, e.g., [26,27]. The character of the spectrum and the shape of each peak are clearly defined by the metric mentioned above; however, in practice, it is often the case that the technicians performing an inspection using resonance methods use their subjective experience to select a particular peak. In this case, the shape of the peak is assessed by the technician’s feelings. Thus, an experienced diagnostician can assess whether the chosen peak matches their experience and notion of a dominant peak. This experience is particularly important when considering highly degraded test bodies where the dominant frequency may not be the one with the highest amplitude. In this case, it is an impetus to identify another metric that would be able to simulate this human factor experience and, in this area, it is advantageous to use a combination of a multi-criteria evaluation algorithm combined with a machine learning model.

The multi-criteria scoring algorithm can assign a score value to a given observation based on the selected weights. The algorithm itself was first published by Saaty [28]. The method was first published in 2021 [29] in the form of using this tool to evaluate frequency peaks. If the evaluated parameters are close to the desired value, the value of the score will be higher. To use this method successfully, the weights of the monitored parameters need to be set. In this case, the weights and setpoints are based on the experience of the technicians and are shown in Table 1.

From these scales, it can be seen that, for example, the value of prominence should reach a maximum value and is more important than amplitude, frequency or width. Conversely, the width of the peak should be as low as possible and is more important than the height of the amplitude or frequency. The frequency value is not very important in this evaluation because different test bodies may have different resonant frequencies depending on their shape, material and internal structure. For the purpose of processing, the natural frequency of the observed peaks was expressed by the relation:(1)$$f_{RL,RT}=\left|\frac{f_{i}}{f_{mT,mL}}-1\right|$$
where:*f_RL,RT_* is the frequency ratio [%],*f_i_* is the frequency of the dominant peak [Hz],*f_mT,mL_* is the average frequency in a given test direction (*f_mt_* = 2000 Hz, *f_ml_* = 4300 Hz).

The average frequencies in a given test direction (*f_mT_* and *f_mL_*) were determined according to masonry elements without defects in the internal structure—durability class 1 (Figure 8).

The variable used for classifying the bricks is, therefore, the relative deviation of the frequency from the specified average SFB frequency in a given testing direction. The resulting value for *f_RL,RT_*, thus expresses how much a given brick differs in frequency from the average of all observed bricks.

In addition to the aforementioned metrics of the first three peaks, score values were added. The findpeaks function was used to select peaks with parameters of a minimum prominence value of 0.03% of the maximum amplitude of the spectrum and a minimum distance between adjacent peaks of 20 Hz. In this way, 100 peaks from each spectrum were separated. Statistical parameters such as mean, standard deviation, peak-to-peak and skewness were then calculated from these peaks. In this way, a total of 20 parameters were obtained from each measured spectrum in the longitudinal and transverse directions in the fully saturated and fully dried state. The frequency parameters were further supplemented by the A-absorption, which was measured at the beginning of the freezing cycles.

To predict the assigned class, the classification toolbox within Matlab software was used. So, specifically, in this case, it is supervised machine learning where there is a set of observations, x, and their classes are known, y. This set is then divided into a learning set and a test set. For large datasets, this split can be performed by random permutation in the ratio of 75:25. Furthermore, a cross-validation algorithm with transfer can be used for this partitioning. The input dataset is divided into subsets. One subset serves as a test set; the remaining subsets serve as training sets. The classifier trains the model on the training set and uses the test set to test the accuracy and performance of the model. This process is repeated several times, each time with a different subset forming the training and test set. In this paper, cross-validation with a transfer of 5 was used. According to the chosen procedure of extracting the features and creating a classification model, it was possible to determine the reliability of the whole methodology in predicting the probable durability (lifetime) of the test set of measured bricks.

To verify the correct identification of defects in the internal structure of the elements and their durability (resistance to repeated freezing and thawing) by non-destructive methods, a destructive frost resistance test was performed. Test elements in the saturated state were cyclically frozen and thawed. One freezing cycle consisted of 16 h of freezing at −20 °C followed by thawing in water for a minimum of 2 h at +20 °C [4]. The resulting number of freeze cycles for each test sample was then recorded.

During the main experiment, sub-factors that affect the durability of the wall elements were also investigated to understand this issue more comprehensively. For example, the effect of humidity on the first natural frequencies and transit times of ultrasonic waves was investigated. The finding of this sub-work was to determine the parabolic dependence of the quantities determined by non-destructive methods on the moisture content of the elements. In many papers, a linear dependence is assumed, but in this case, significant errors can be made. Results from this sub-experiment were published in [30]. Another sub-research was to monitor the changes in first natural frequencies and transit times as a function of the number of freeze cycles. In the framework of this work, three possible types of sample defects were determined [18].

## 3. Results

From initial measurements made on site, it was found that the transit time of ultrasonic waves could not be used as a satisfactory metric for assessing likely durability (resistance to freezing cycles). Figure 9 shows the relationship between the resonant frequency and the velocity of longitudinal ultrasonic waves (p-wave) measured per length of brick in the dried state. The colour shows the number of cycles that each brick lasted before destruction. A dominant diagonal can be observed in the data, which confirms that these observed parameters are correlated; however, in terms of the distribution of bricks that have lasted 25 cycles or more, there is significant mixing with bricks that have already disintegrated after 5–10 cycles. At the same time, a non-negligible number of dependencies can be observed that lie outside the main diagonal, which could be called anomalies. Looking at a similar dependence of water absorption on ultrasound velocity in Figure 10, it can be seen that the average ultrasound velocity lies in the range of 1600–3000 m^−s−1^ and the bricks are mixed throughout the observed range of freezing cycles.

On the other hand, the majority of the measured bricks had a value of absorption *A* ranging from 12–24%, with an insignificant correlation with ultrasonic velocity, as shown in Figure 11. For the same value of water absorption, there is a brick with both the highest and the lowest durability (lifetime) expressed in the number of freeze cycles. Thus, it can be concluded that the probable lifetime of the brick under consideration cannot be reliably predicted from common parameters such as dominant frequency, ultrasonic velocity, or absorption rate.

Bricks were divided into different classes based on their durability interpreted by a count of F-T cycles, which the bricks withstand in the automatic freezer at set conditions described in Section 2 Materials and Methods. Each class is shown in Figure 12.

Moving on to the results of the feature extraction algorithm, the application of the multicriteria method to the frequency spectrum peaks will be first up. Statistical comparison in terms of probability of the observed parameters’ frequency, peak width amplitude, peak prominence and the corresponding score is shown in Figure 13. This is a total of 4136 observed peaks, which were selected by the findpeaks function from 167 frequency spectra of all 41 measured bricks.

The amplitude is spaced between −9 and −6 dB. Classes with a higher freeze cycle life generally have higher amplitude, and conversely, classes with lower life have lower amplitude values. However, it is in the middle of the histogram that these parameters overlap. The same is true for the frequency ratio *f_R_*. Perhaps the biggest difference occurs in the case of peak prominence, where each class’s peak probability represents a different part of the prominence variance from −4 to −15 dB. Although the classes were designed according to the increasing number of cycles, the order exhibited by the prominence value shows that class 1 is very similar to class 3 and class 4 is very similar to class 2. From this perspective, it can be said that this is the effect of a smaller statistical sample of bricks that would otherwise show a lower prominence value with a higher class.

For a spatial, graphic representation of these quantities, see Figure 14. There are three main point clouds A, B and the largest one C. Clouds A and B are clearly separated by a gap due to the score value, and if only these points were used, there would be relatively high confidence in classifying the bricks. Clouds A and B consist exclusively of classes 2 and 4, which, moreover, supports the previous finding that classes 2 and 4 are very similar. However, these clouds reach a maximum peak width of 13 Hz, which is especially true for extremely narrow and small peaks. In contrast, the peaks located in cloud C are interpenetrating and composed of all the remaining measured peaks, and all classes are equally represented. Thus, this observation invites the possibility of a multi-stage assessment for the presence of peaks from Cloud A and Cloud B, and then, if peaks from these clouds were not found, a more detailed method would need to be applied.

A more detailed assessment is focused on obtaining the selected 20 parameters and building a classification model based on the principle of an ensemble of classifiers [31]. The statistical comparison of selected parameters is shown in Figure 15.

In the total number of selected parameters, variants such as Freq1, Freq2, etc., are present; in this case, these are the parameters of the dominant peaks in descending order of prominence. The parameters Freq1 or Width1 describe the same peak but in different ways. The statistical parameters are:The Kurt and Skew parameters describe the tailedness and skewness of the score distribution of selected peaks from each assessed spectrum,MeanScore expresses the average score of the peaks,ModeWidth describes the modus of the peak width,StdScore expresses the standard deviation of the selected peaks,TotalScore expresses the total sum of the scores of the selected peaks.

Figure 15 shows dependencies that are difficult for humans to understand, and even using correlation diagrams they are not a good tool for interpreting this type of data. Since this is multidimensional data, which has a total of 18 dimensions, it is appropriate to assess the individual dependencies using the success of the classification model and the resulting decision diagram. To understand more about the dependence between the parameters, this must be done, in part, by expressing the dependence between the chosen parameters Freq1 and the absorbance and standard deviation of the peak scores. For an example of such a representation, see Figure 16.

These plots in Figure 15 illustrate the probability of each observed variable within the total range of the variable, where classes 1 ÷ 4 are represented by different colours. In some parameters, the probability is very similar among all classes, such as the Frequency ratio of the second highest peak *f_Δ_*_2_. Parameters such as Sum of peak score *ΣS* and peak count shows a significant difference in probability between class 1, 3 and 2, 4. From all of these parameters, the most variance occurs in the first highest peak *f_Δ_*_1_, the score of peak *S*_1_ and the prominence of dominant peak *P_dom_*_1_. The parameters with the highest variance are the most suitable for classification model generation.

It is evident that approximately 60% of the bricks, which have the highest class and, therefore, can withstand the least number of cycles, cluster around one centre in both cases. The rest of the observations of this class are spread evenly among the other points of classes 1, 2 and 3.

It is similar in the case of the ratio between the frequency ratio and the standard deviation of the frequency peaks. Here it can be seen that class 1 achieves the lowest frequency ratio value, which means that its dominant frequency is closest to the average SFB frequency *(f_mt_ =* 2000 Hz, *f_ml_ =* 4300 Hz). At the same time, Class 2 and Class 3 observations and some Class 4 observations are also located in this area.

Thus, from this perspective, it can be shown that it is not possible to successfully classify SFB by classical methods and evaluation because it involves the complex behaviour of several different parameters. Thus, if this classification task is carried out by the algorithm mentioned in [31], it gives a decision as to how and in which situation the proposed classification model is to be used.

Using only a classification model with the ability to distinguish between class 2 and class 4, a fairly satisfactory reliability of 88% for class 2 and 72% for class 4 is obtained. Parameters include the resonant frequency in the fully saturated and dried state, the number of peaks, the difference between the frequency in the fully saturated and dried state, the number of peaks, the width and Poisson’s ratio, and the absorbance were used for this model. Model 1 is illustrated using the confusion matrix in Figure 17.

The resulting model success rate corresponds to the results in Figure 14 and indicates that the two classes are close but can be successfully recognised. From a practical point of view, however, this procedure requires knowledge of absorption rate A, which requires the selected brick to be fully saturated with water and then dried to a stable weight. This procedure is time-consuming and cannot be carried out on bricks that are already in place.

If the parameters described in Figure 15 are used without including the absorption, it results in 25% for class 1, 56% for class 2 and 69% for class 4. The success of Model 2 is shown in Figure 18. In this case, it turns out that both classes 2 and 4 are very close to each other. However, in this case, there is a misclassification of Class 2 bricks into Class 1, 3 and 4 and a misclassification of Class 4 bricks into Class 1, 2 and 3. From this point of view, the model can be used on embedded bricks but has very little reliability.

The last and the most successful model (Figure 19), number 3, uses the parameters used for model 2, enriched with the knowledge of absorbance A, with an overall success rate of 85%. From this perspective, it is the most accurate model so far, achieving 83% accuracy for Class 1, 88% for Class 2, 75% for Class 3 and 86% for Class 4. The model was used from observations of the frequency spectra of both the saturated state and the dried state in both longitudinal and torsional directions. From this point of view, it does not matter whether it is used on a dried or saturated sample.

The resulting decision tree for classification is shown in Figure 20. The decision tree is a graph that uses structure of nodes, branches and leaves. Each node is made by binary condition which leads to either a node or a leaf. In this setup a leaf is predicted class. Each node is formed by assessed features. The presented decision tree in Figure 20 was generated by machine learning algorithm, and so are the values of binary condition for each node. This tree clearly shows the importance of absorbance for classification; it is by far the most common parameter used by the model here. The second most common is the parameter Freq1 or the frequency ratio of the first peak. There are also statistical parameters such as kurtosis, skewness, number of peaks or average score of frequency peaks.

The analysis makes it possible to determine which state and type of resonance measurement are the most suitable for obtaining the highest classification accuracy for both model 2 and model 3. This comparison is shown in Table 2.

Comparing the success rate of the model, it can be seen that the saturated state in the longitudinal testing direction achieves a success rate of 90.48%. The lowest success rate is 78.05% for the dried state in the transverse testing direction. In the case of model 2, the longitudinal direction in the dry state achieves the highest success rate—69.05%—and the longitudinal direction in the saturated state has the lowest success rate—42.86%.

## 4. Discussion

For the possibility of “upcycling” historic bricks, for both new buildings and the renovation of historic buildings, it is crucial to choose a durable material that fulfils both a functional and an aesthetic role. The requirements for historic bricks reintegrated into the structure should be the same as for newly manufactured bricks. The results of this work have shown that by using a non-destructive resonance method and the known absorption of solid fired bricks, it is possible to predict their durability (service life) with relatively good accuracy, without failure. The classification tree (Figure 20) demonstrates that absorbability is a significant factor in evaluating the durability of solid-fired bricks. It is a non-linear characteristic that cannot be described by a simple correlation. In the case of the results obtained from the resonance method, a success rate of 85% (Figure 19) was achieved. The disadvantage of this model is the need to know the absorption of the masonry elements. Therefore, this model is difficult to implement in situ and is more suitable for laboratory testing. The ultrasonic pulse method used did not yield significant results already during the initial on-site testing of the methods. This assumption has also been confirmed by laboratory tests and thus, the method is not suitable for the purpose of this paper. The measurement of ultrasound wave propagating through the element provides fewer parameters than the resonance method, where a whole spectrum is a result, and we can observe many different metrics and features. The whole procedure was verified on a selected sample of 41 bricks, which, by their time, age and quality, cover a wide historical and material spectrum. Therefore, it can be stated with some confidence that, within the framework of this study, it is possible to propose a classification model that can predict the probable durability (service life) of a historic brick based on NDT measurements. To bring the method into practice, a similar algorithm will be converted into Python using the SciPy and NumPy libraries, which are commonly used for this type of task. At the same time, the learning dataset will be enlarged to cover more sources of brick measurements. In fact, there was an uneven qualitative representation of bricks in the tested set. Specifically, Class 2 and 4 were represented in greater numbers than Class 1 and 3. Therefore, more class 1 and 3 bricks need to be included so that the entire dataset becomes balanced. Thanks to the current level of microphones currently used in consumer electronic devices, which can be used for the recording of resonance signals of elements tested by the resonance method, we can assume that the proposed classification procedure and algorithm could be used by mobile devices, such as smart phones, or tablets.

## Figures and Tables

**Figure 1 materials-15-05882-f001:**
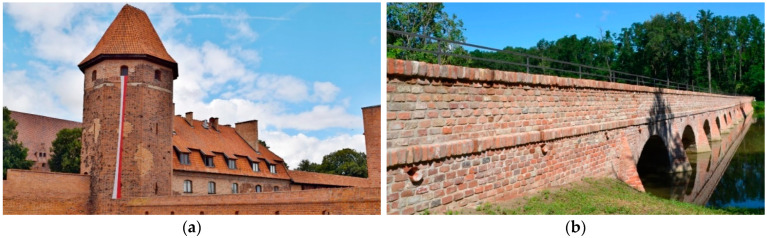
(**a**) Reconstruction of the Gothic castle in Malbork (Poland) with modern masonry elements, (**b**) Reconstruction of a baroque brick bridge near Mikulov (Czech Republic) with historical masonry elements.

**Figure 2 materials-15-05882-f002:**
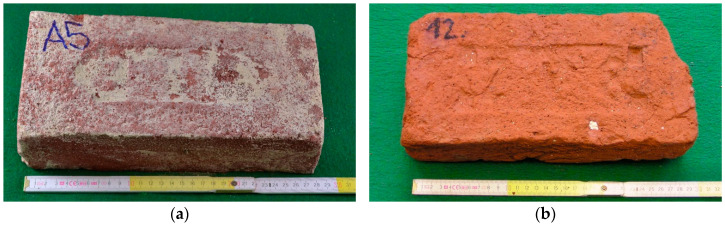
(**a**) historical solid brick—high firing temperature, (**b**) historical full brick—low firing temperature.

**Figure 3 materials-15-05882-f003:**
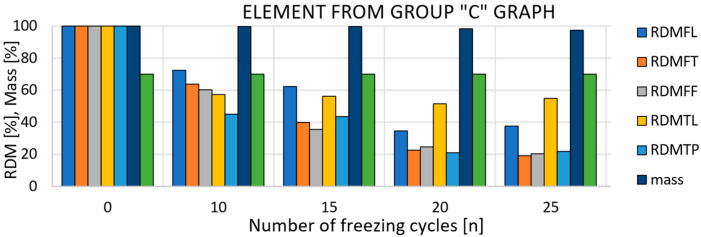
Example of a solid-fired brick of type 3 damage [18].

**Figure 4 materials-15-05882-f004:**
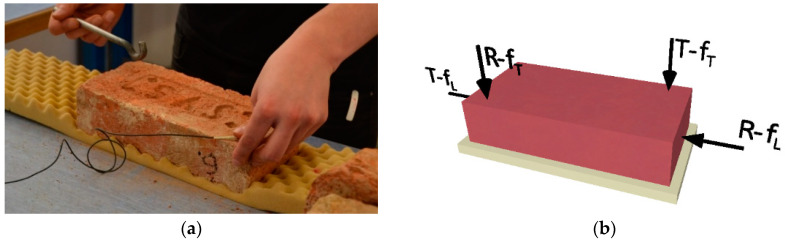
(**a**) Example of measuring the first natural frequencies of torsional vibration (f_T_), (**b**) Arrangement of sensors (“R”) and exciters (“T”) for longitudinal (f_L_) and torsional (f_T_) oscillations.

**Figure 5 materials-15-05882-f005:**
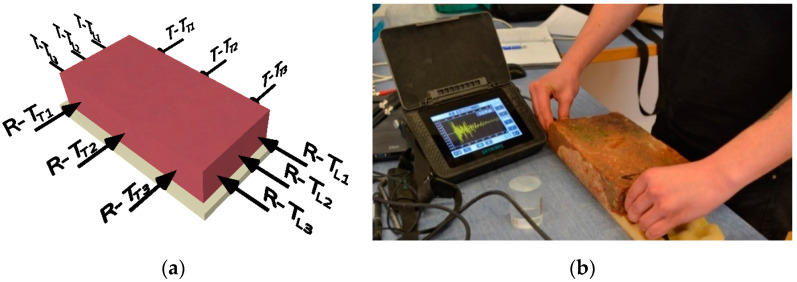
(**a**) Positioning of the transducer (“R”) and exciter (“T”) to determine the transit times in the longitudinal (TL) and transverse (TT) directions (always in 3 lines), (**b**) Example of a longitudinal travel time (TL) measurement.

**Figure 6 materials-15-05882-f006:**
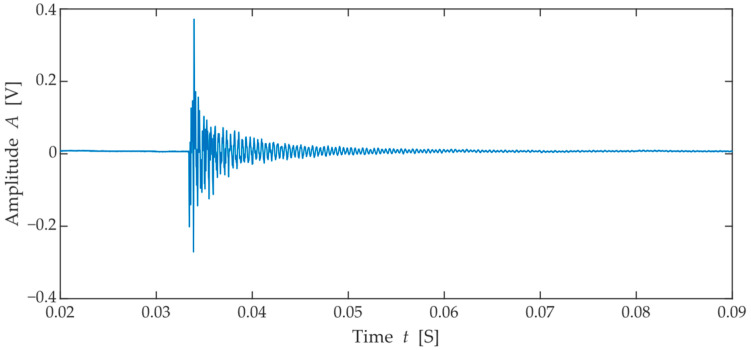
Illustration of acoustic resonance signal: Signal in the time domain.

**Figure 7 materials-15-05882-f007:**
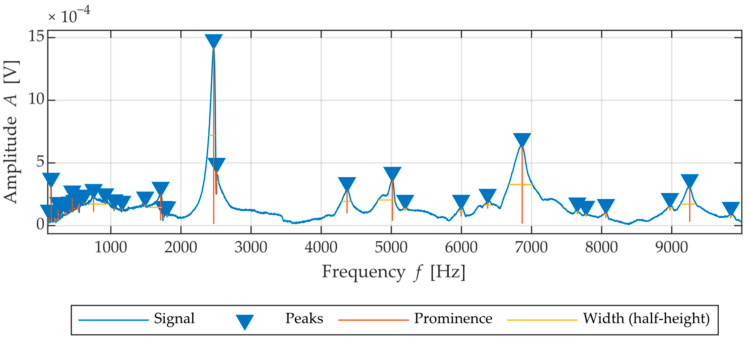
Illustration of acoustic resonance signal: Extracted frequency spectrum with highlighted peaks with various parameters such as frequency, amplitude, width and prominence.

**Figure 8 materials-15-05882-f008:**
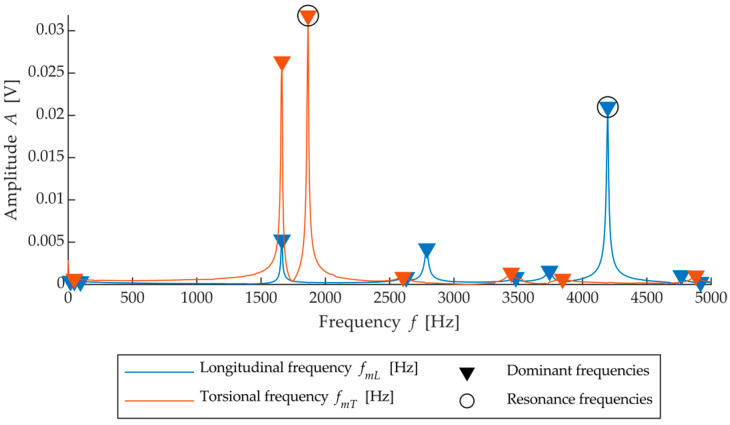
The frequency spectrum of a masonry element without defects in the internal structure.

**Figure 9 materials-15-05882-f009:**
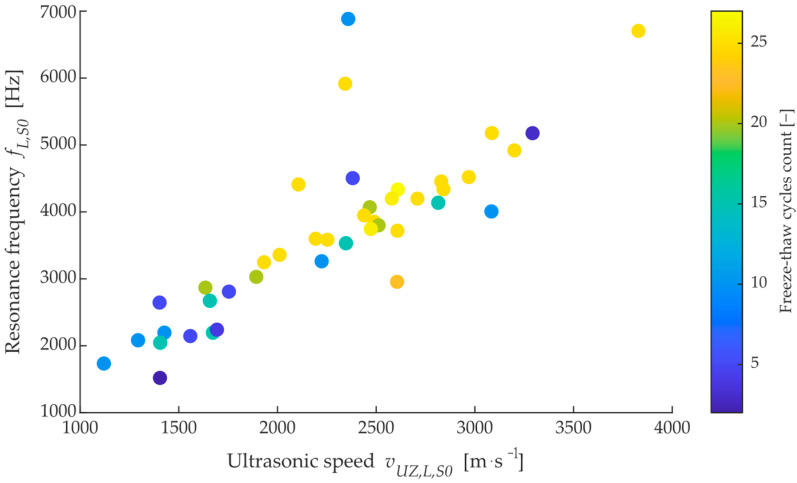
Dependency between longitudinal resonance frequency on ultrasonic velocity (p-wave) in longitudinal axis also in a dried state.

**Figure 10 materials-15-05882-f010:**
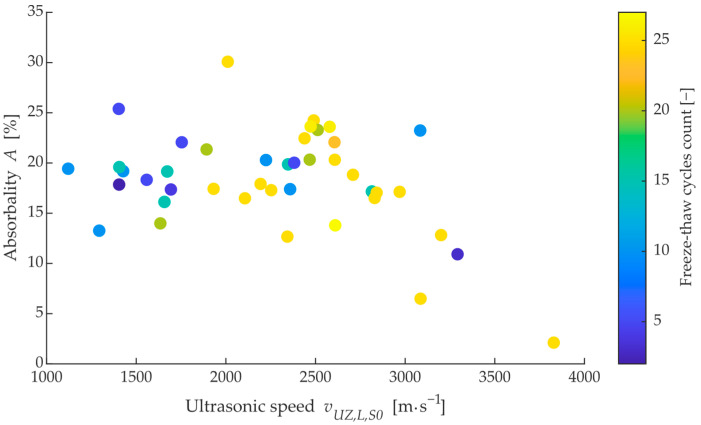
Dependency between water absorbance on ultrasonic velocity (p-wave) in longitudinal axis also in a dried state.

**Figure 11 materials-15-05882-f011:**
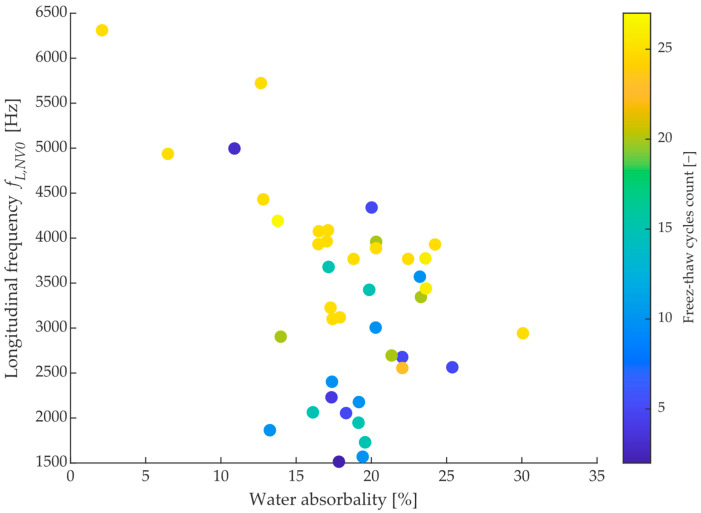
Dependency of longitudinal frequency in the saturated state on measured water absorbability with highlighted freeze-thaw cycles, which each brick endured up to destruction.

**Figure 12 materials-15-05882-f012:**
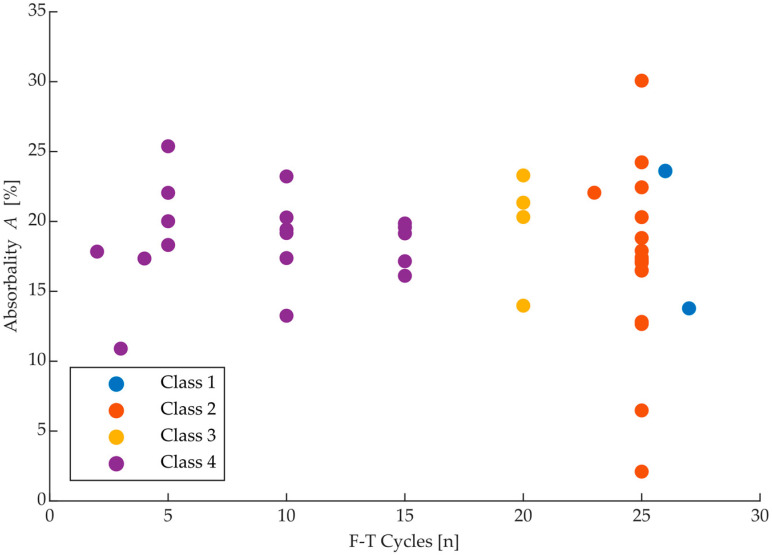
Comparison of a number of F-T cycles and absorbability of bricks with their set classes.

**Figure 13 materials-15-05882-f013:**
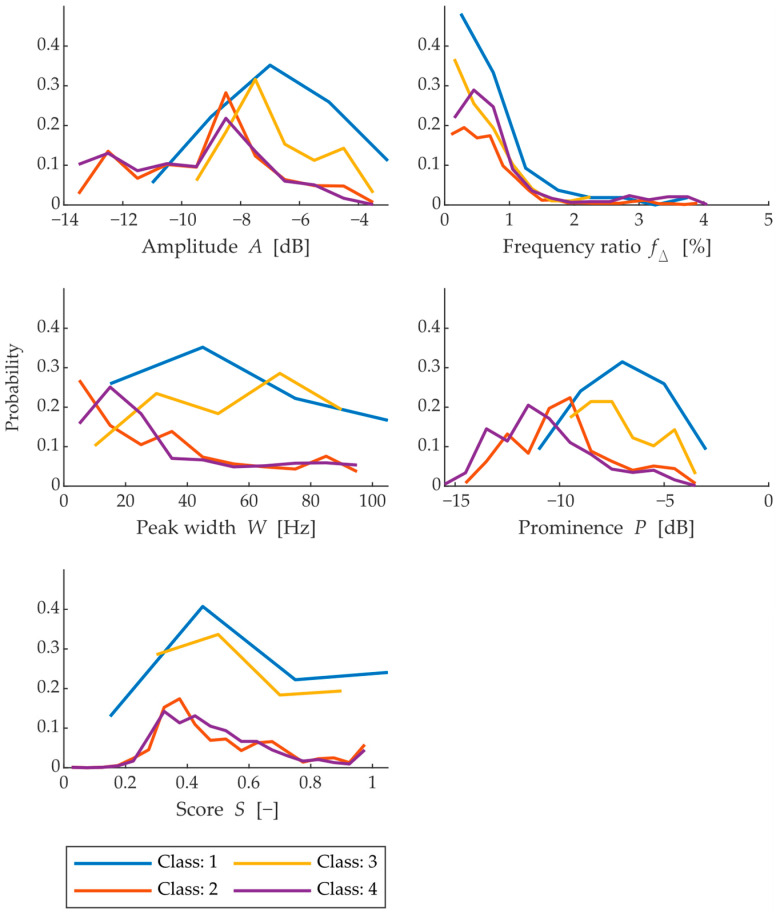
Histogram of peak parameters across all measured frequency spectrums (for peaks with width < 100 Hz).

**Figure 14 materials-15-05882-f014:**
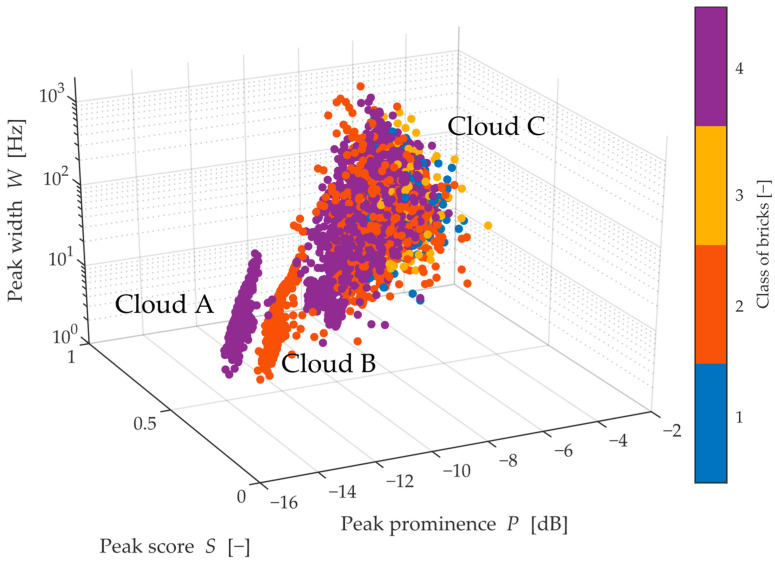
Distribution of peak metrics with a comparison with classes and score of individual peaks.

**Figure 15 materials-15-05882-f015:**
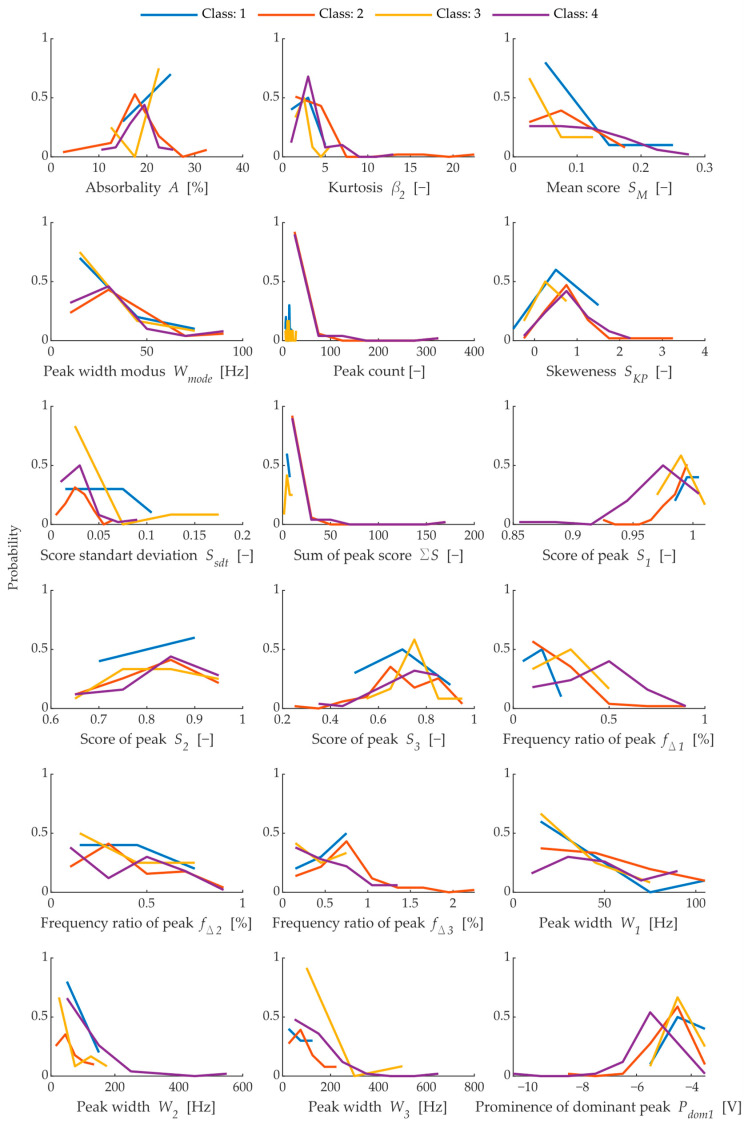
Comparison of all extracted features from the frequency spectrums and properties of fired bricks (absorbability).

**Figure 16 materials-15-05882-f016:**
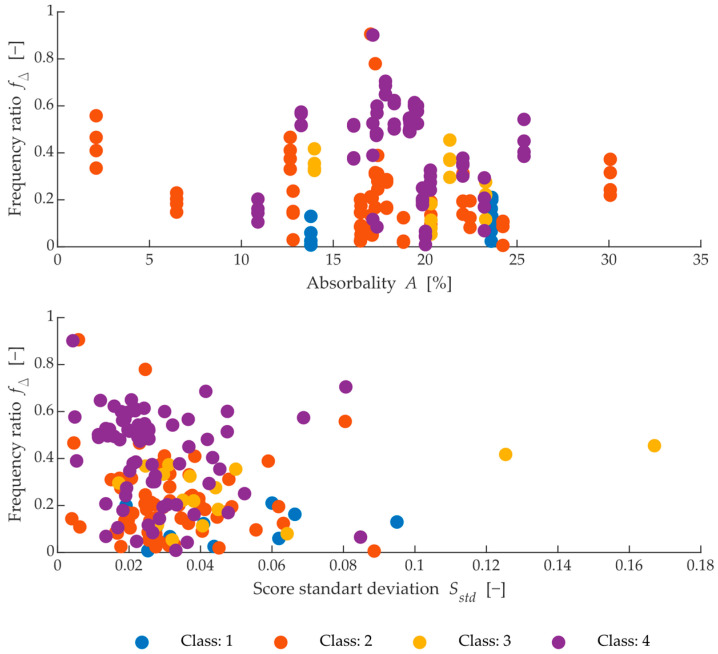
Comparison of absorbability, mean score value (**upper**), prominence of dominant peak and skewness (**bottom**).

**Figure 17 materials-15-05882-f017:**
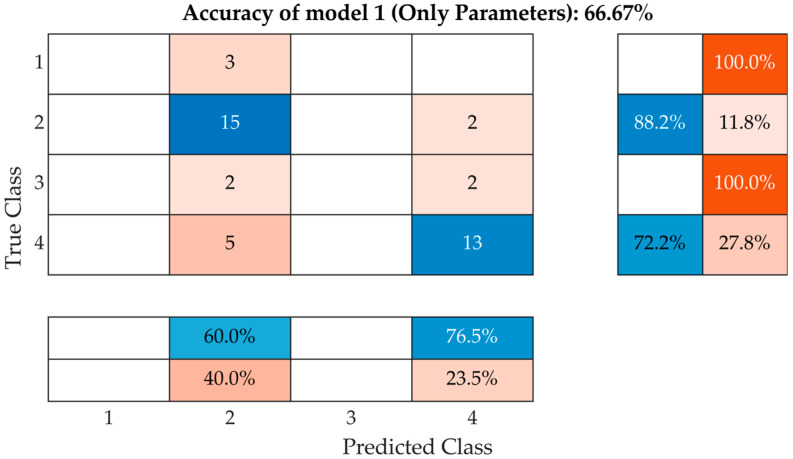
A model with parameters extracted from *f_L_* without statistical parameters and score values.

**Figure 18 materials-15-05882-f018:**
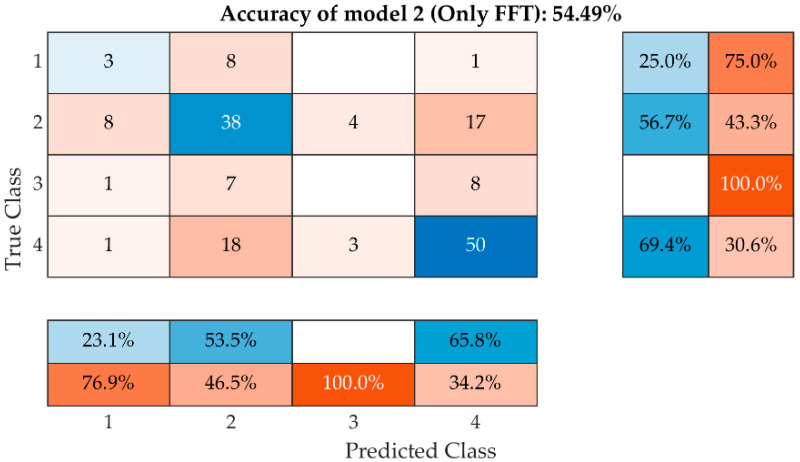
A model with only FFT parameters is known without absorbability.

**Figure 19 materials-15-05882-f019:**
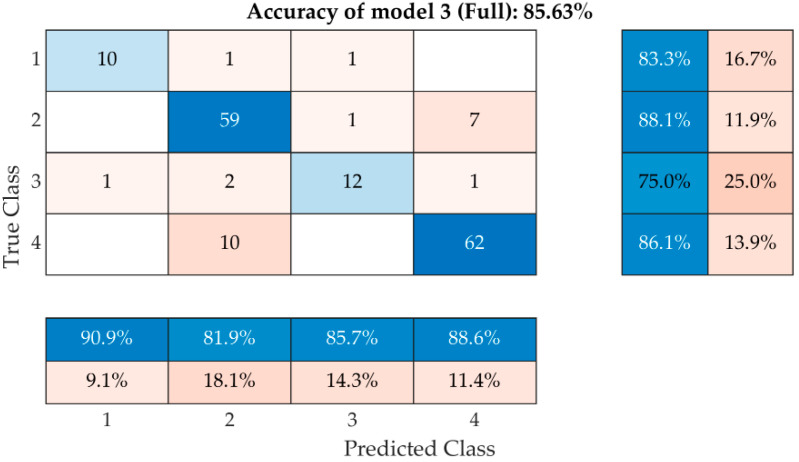
Confusion matrix of the most effective model with all known parameters.

**Figure 20 materials-15-05882-f020:**
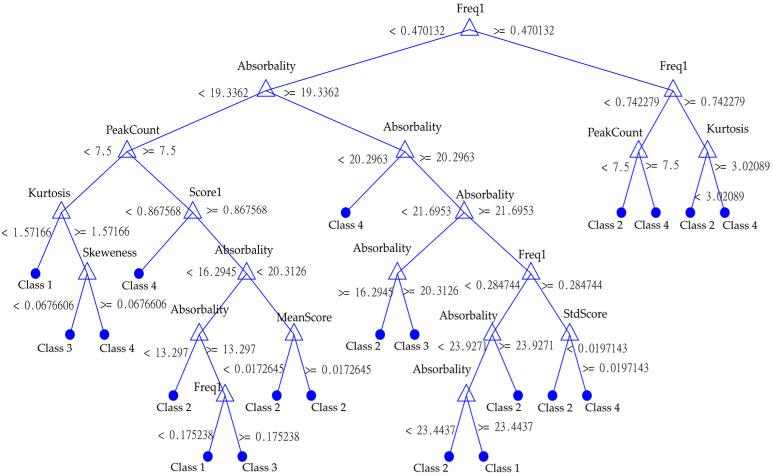
Example of the designed model’s classification tree with parameters, conditions and resulting classes.

**Table 1 materials-15-05882-t001:** Weights set for scoring frequency peaks (A-amplitude, F-frequency, P-prominence, W-width).

Param.	A	F	W	P	Demand
A	1.0	5.0	1.0	0.2	Max
F	0.2	1.0	0.2	0.1	Max
W	3.0	5.0	1.0	0.3	Min
P	5.0	10.0	3.0	1.0	Max

**Table 2 materials-15-05882-t002:** Accuracy of model 3 in all combinations of dried or saturated specimens and testing by longitudinal or transverse waves.

Type of IE Testing	Model 2 Accuracy in Different Groups [%]		Model 3 Accuracy in Different Groups [%]	
	S0—Dried	NV0—Saturated	S0—Dried	NV0—Saturated
*f_L_*—longitudinal	69.05	42.86	85.71	90.48
*f_t_*—transverse	56.10	52.38	78.05	88.10

## Data Availability

Data supporting the reported results are included in the Appendix A of this article. These are resonance measurement data in csv format.

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
