# Peer review of "Predicting the Durability of Solid Fired Bricks Using NDT Electroacoustic Methods"

_materials, 2022, doi:10.3390/ma15175882_

Round 1

Reviewer 1 Report

Subject:  Corrections and comments made to paper entitled: “Predicting the durability of solid fired bricks using NDT methods” (Manuscript Number: Materials-1842776)

Dear Madam/Sir,

Please find below the corrections and comments made to the paper above.

Introduction

Page 5, lines 155 to 159

Text: “At the same time as the non-destructive measurement (monitoring the development of defects in the internal structure) was monitored, the development of defects visible on the surface of the elements was also monitored. The development of defects visible on the surface was expressed as a percentage of weight loss due to chipping from the elements.

Comment: There are repetitions in the same sentence or from one sentence to another. I suggest a modification for a better understanding to the reader.

Materials and methods

Figure 6 and 7

Comment: Since the images are related, I suggest a combination of the two figures into one.

Figure 8 and 9

Comment: Since the images are related, I suggest a combination of the two figures into one.

Results

Figure 17 and 19

Comment: For a better comparison, can the authors keep the same scale in the Y axis, since in all of the graphs the values represent the probability.

Conclusions

Comment: Only two NDT were used the authors do not give a suggestion of there performance, like the advantage or disadvantage of using them, which seems to the best amount them. The absorption did not shown a good correlation in the results, so it cannot be used as the authors mention as a method to predict the durability as stated in line 560 and 561.

Author Response

On behalf of all the authors, I would like to thank you for your helpful comments on this paper, which made it .

Reviewer 2 Report

After going through the manuscript, I recommend Minor Revision for this manuscript. The topic and examples are interesting and suitable for this journal. Following questions are for the authors' reference.

 1. Some backgrounds and simple works that are not closely related to the innovative work of this paper can be simplified.

 2. In this paper, 41 sample bricks are selected for structural measurement. Why not choose more samples? From the field situation, it is not difficult to obtain more samples.

3. Line 356, how do you get these two key data? fmT and fmL.

 4. From Figure 17 to Figure 20, the difference between the four groups is not obvious. Is this classification method not clear?

 5. The last section should be Conclusion. The third section should be Results and discussion.

Author Response

On behalf of all the authors, I would like to thank you for your helpful comments on this article, which have improved the quality of the article.

Reviewer 3 Report

This paper proposes to predict the durability of solid fired bricks to be used in heritage rehabilitation by using non-destructive electroacoustic methods. The subject is of great interest as these techniques are quite available and affordable and allow in-situ measurements which is important in the field of cultural heritage.

My comments and suggestions for this work are the following:

-          The title should include “non-destructive electroacoustic methods” and indicate in a concise but clear way that this work refers specifically to the statistical analysis of experimental results to obtain the predictive methodology.

-          The introduction is very long. The background is widely presented, but the specific subject of the paper should be better specified in the last paragraph of this section, as indicated for the title.

-          The equipment used for the resonant pulse method is not specified.

-          Figures 6 to 9 may be merged in two figures: both pictures in one and both schemes in the other or picture and image corresponding to the same test in the same figure.

-          From Page 7, line 236 to page 8, line 271. Previous results are described in detail. I would suggest to include only a brief summary without figures and give the reference 23 for details. If this reference does not include all the details, then the results should be included in the “Results” section of the paper.

-          Page 9, line 294. The conditions used in the freeze-thaw cycles should be specified at this point.

-          Page 10, line 27. It should be specified what is meant by “IE” methods.

-          Figure 19 – figure caption is incomplete.

-          The results section is difficult to follow for me and I think that for any reader not experienced in statistics. Some short comments could be added to help understanding the figures. In any case, the general ideas obtained from the results are clearly stated.

Author Response

(The authors gave the same response as above.)

Reviewer 4 Report

This article introduces a non-destructive testing method to study the durability properties of solid fired bricks. The article is innovative to a certain extent, and a large number of indoor experiments have also been carried out. However, there are still several problems that need to be improved.

Question1: Pages 1-3- The introductory section describes the need for substantial protection of solid fired bricks, which should be streamlined. In addition, methods taken by other researchers to improve the durability of solid sintered bricks and detect damage should be added to improve the scientific of the paper.

Question2: Page 7- The method of non-destructive testing is widely used in bridges, buildings and other scenarios. The article should reflect whether the non-destructive testing method used is innovative and state the difference from other non-destructive testing methods.

Question3: Pages 6-11- Materials and test methods should be written separately and adjusted in terms of thesis structure. The mechanical strength test method and the freeze-thaw cycle test method should be reflected.

Question4: Page 14 Line 499-  Figure 17 is blurry and should be revised.

Question5: I read the whole article, however, I didn't find the purpose of the article, a lot of content is related to NDT. The article lacks findings on the durability of solid fired bricks and should be supplemented.

Author Response

(The authors gave the same response as above.)

Round 2

Reviewer 4 Report

All questions have been answered or corrected.